# The Optimization of Mechanochemical Processes toward Functional Nanocomposite Materials

Mamoru Senna 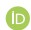

Faculty of Science and Technology, Keio University, Yokohama 223-8522, Japan; senna@applc.keio.ac.jp;
Tel.: +81-90-9293-2758

**Abstract:** Mechanochemical technology is developing rapidly, judging by the scientific information in both basic and applied studies. However, many issues and points of view remain to be discussed. This review presents some new key issues for the optimization of mechanochemical processes in terms of theoretical and practical aspects. Emphasis is placed on powder technology aspects, which are not always discussed compared to functional or microscopic viewpoints. The transfer of chemical species across the interparticle interface between dissimilar species during the mechanosynthesis of nanocomposites offers many new opportunities. Since almost all material transport is preceded by charge transfer, its driving force has been sought using terminology beyond the well-established electrochemical terms. In particular, the valence state of the cationic species involved is of importance. The role of organic compounds throughout the process is emphasized, regardless of their survival in the final product. The similarity with pharmaceutical phenomena is pointed out, although its mentality is very different from that of the synthesis of nanocomposites. The rational amorphization and stabilization of molecular dispersion states with the participation of excipients are discussed. The effects of liquids, either added or formed by mechanochemical auto-liquefaction, are presented with reference to the comparison between wet and dry grinding. The mechanisms of the apparent stabilization of the mechanically activated states of the products are elucidated to investigate the practical applicability of these mechanochemically synthesized products. Finally, the most important aspects for the optimization of the mechanochemical processes of functional nanocomposites are listed.

**Keywords:** precision mechanosynthesis; interparticle materials transfer; molecular dispersion; auto-liquefaction; nanoglassy states

## 1. Introduction

The potential of mechanochemistry, both in basic research and technological application, is rapidly expanding [1–4]. With a deeper understanding of the associated mechanisms [5–7], concepts of precision mechanosynthesis are growing in many technical genres. Organic synthesis with controlled chirality [8], or the inclusion of high-entropy compounds containing a large number of cationic species [9–11], has been accelerated by the participation of organic chemists or metal scientists in conjunction with mechanochemistry, rising in popularity. Materials with even more complicated structures, such as double perovskites [12–14] or metal organic frameworks (MOFs) [15–19], have been also mechanosynthesized. Very fine structural details, such as controlling the anti-site disorder in double perovskites, have also been challenged [20,21]. Increasing interest in downstream mechanochemistry toward industrial application is in line with greener chemical processes, and hence with the UN-led SDGs [22–24]. Representative effects are found in industry, typically in pharmaceuticals [3,25–27].

The vast majority of raw materials and end products are in the form of particulate solids. However, related discussions in materials science are not always based on the viewpoint of powder technology. This review focuses on the stress-induced physicochemical phenomena at interparticle boundaries when a powder mixture is stressed. They are based

on the transport of chemical species across the interparticle interface. This occurs from the species with the lightest mass, electrons, and protons and proceeds to heavier ionic species. The charge transfer phenomena involving solid organic species are relatively less studied, although they may play a crucial role in the synthesis of complex metal oxides [28,29]. Precision control of the microstructure of the mechanosynthesized products is also achieved by the proper choice of starting materials, in conjunction with the fine redox reactions that take place within a reaction mixture under mechanical stress.

In pharmaceuticals, the better dissolution and absorption of the active pharmaceutical ingredient (API) is always explored for a higher bioavailability with the controlled release of the API. Many attempts have been made to amorphize the API under mechanical stress [30,31]. The chemical interaction at the organic–inorganic grain boundaries can be used not only for the rational amorphization of the API, but also for the stabilization of molecular dispersion states. Pharmaceutical technology may seem far removed from materials synthesis, however, there are many similarities when it comes to the chemical interaction across the inorganic–organic particle interface under mechanical stress.

The effects of either added liquids or mechanochemical auto-liquefaction are another new perspective. A change in the mechanical properties of metals with surfactants belongs to the traditional findings under the concept of Rehbinder effects [32]. The mechanisms of the apparent stabilization of the mechanically activated states of products are also being elucidated. Solid-state chemists know that, when powdered materials are introduced into a grinding mill, not only brittle fragmentation, but also inelastic, irreversible structural changes occur, accompanied by the formation of lattice defects [33–36]. These irreversible non-geometrical factors are associated with mechanochemical phenomena. However, these mechanochemical phenomena are now far beyond classical crystallography. This is not only due to the assumption of amorphous or glassy states. Single-molecule mechanochemistry is one of the topics beyond the scope of classical concepts [4,37,38].

Despite the remarkable progress in mechanosynthesis at the laboratory scale mentioned above, we still face several difficulties in bringing such technology to be applied at the industrial level. Just to address this problem, a new action was launched in 2019, funded by the European Cooperation in Science and Technology (COST) as Action: CA18112—Mechanochemistry for Sustainable Industry (MechSustInd) [39]. This pan-European action, which has now been extended to a global scale, has a great dynamism, especially in the field of organic synthesis-based pharmaceutical industry. Scaling up from gram to tonnage orders is sensational [40–43]. Scalability and affordability are particularly important, e.g., the mechanochemistry of cellulose [44].

The objective of this review is to offer a guideline for selecting preferential items for the various goals of mechanochemical operations, i.e., (a) new bridging bond formation by charge transfer; (b) the precision control of lattice order and imperfection; and (c) sustainability and affordability.

## 2. Diversity of Mechanisms in Mechanochemistry

In recent decades, the mechanistic issues of mechanochemistry have been extensively developed. One of the major issues is the mentality of organic chemistry in the interest of the rationalization of organic synthesis procedures [17,24,45–48]. Most of this outstanding work has been conducted by organic chemists. Although the conventional boundary between organic and inorganic chemistry is becoming more and more diffuse, differences in the mental background still remain. Therefore, it is extremely important to study the changes in the stability and state of migration of the chemical species on the surface of each society when dissimilar particles are brought into contact and an external force is applied.

The migration of charged species is ideally dominated by the electric potential field. One of the oldest related findings of charge transfer across solid interfaces is contact electrification (CE). Phenomenologically, CE has long been known, but its mechanisms remain controversial. A recent study on a system of carbon and silicon dioxide showed that CE is partly driven by the surface-dipole-induced potential during contact [49]. They

further discussed the existence of a separation-dependent potential barrier at the contact interface. It has also been pointed out that CE is closely related to mechano-luminescence, i.e., the light-emitting behavior of materials upon the application of mechanical stimuli [50]. Strain-induced CE has also been reported [51]. All these reports unequivocally indicate that the charge transfer across solid interfaces under mechanical stress is associated with the fundamental mechanisms of CE.

Sophisticated discussions have been had about electrochemistry. The actual electrochemical charge transfer is significantly influenced by overpotential, whose components are very complicated. This is studied especially in the interest of all-solid rechargeable batteries [52–54]. These works discussed charge transfer in an electrochemical context, forcing the question of how to avoid obstacles against the smooth transfer of charge-bearing species, mostly $Li^+$, across the electrolyte–electrode interface. Plastic deformation plays an important role in interfacial charge transfer, but is mostly ignored in such an electrochemical discussion [55].

Under mechanical stress, an additional component appears due to local deformation and associated lattice imperfections [56,57]. Plastic deformation is irreversible if it is highly localized, so the charge transfer is inevitably affected by the resulting change in electron density distribution. Recognizing this is particularly important in mechanochemical processes, since the electrical driving force of charged species is dominated by the local electrical potential gradient and hence by the local electron density distribution. Lee et al. discussed this point in detail using in situ compression neutron diffraction (ND) measurements [58].

A comprehensive review of mechanochemistry with guiding terminology has been published, starting from a hierarchical representation of the main effects of mechanical action on different systems, ranging from single molecules to multiphase solid powder mixtures [5]. The basic mechanisms of mechanochemical phenomena are also observed from the point of view of different channels of stress field relaxation [2]. In fact, relaxation takes place during periodic stressing by different milling devices [3]. These parallel phenomena cannot be expressed by a linear equation because they can be interpreted as multiple overlapping localized reactions [59].

The coexistence of strong intramolecular covalent bonds and weak non-covalent intermolecular interactions characterizes the mechanochemical phenomena in organic molecular crystals [2]. Another important concept in organic synthesis is chirality. In pharmaceutically active compounds, an enantiomerically pure form is of vital importance to avoid the coexistence of an undesired or ineffective enantiomer, distomer with the desired one, and eutomer [60]. The control of chirality related to mechanochemistry has been studied [61]. It is noteworthy that the control of the enantiomer is associated with near-equilibrium deracemization. The latter is related to a classical concept of solution chemistry, Ostwald ripening [62]. The control of Ostwald ripening can be controlled by grinding, a primitive action of mechanochemistry [63].

## 3. Nanoglassy and Nanocrystalline States of Complex Oxides

Mechanochemical technology is developed from the formation of new complex oxides visible through conventional X-ray diffraction. More light is shed on the finer control of nanostructures. A typical example is the introduction of the concept of nanoglasses, where a solid consists of nanometer-sized glassy regions connected by interfaces with reduced density. Such glassy states can exist at room temperature due to the presence of boundaries around which nanoglass clusters are dispersed [64]. This will be mentioned later in the context of stabilization. Since nanoglass materials have been primarily associated with alloys, preparative methods have also been used in alloy preparation, such as inert gas condensation [65] or arc melting [66]. Since mechanochemical alloying, or mechanical alloying, has been found to be versatile for the preparation of bulk amorphous alloys [67,68], it is natural to prepare nanoglass materials via mechanochemical routes as well. The present author, together with his colleagues, has demonstrated some examples of this possibility [69].

In a case study discussed below, a stoichiometric mixture for pyroxene $LiFeSi_2O_6$, consisting of a-$Fe_2O_3$, $Li_2SiO_3$, and $SiO_2$, was ground together. Mechanosynthesis proceeds with simultaneous amorphization after grinding for 13 h, as shown in Figure 1a [69]. By stopping the mechanochemical treatment for only a short time, 0.5 h, then heating at 1000 °C in a thermal analyzer at a heating rate of 10K/min and then cooling the furnace, we obtained well-crystallized pyroxene $LiFeSi_2O_6$, as shown in Figure 1b.

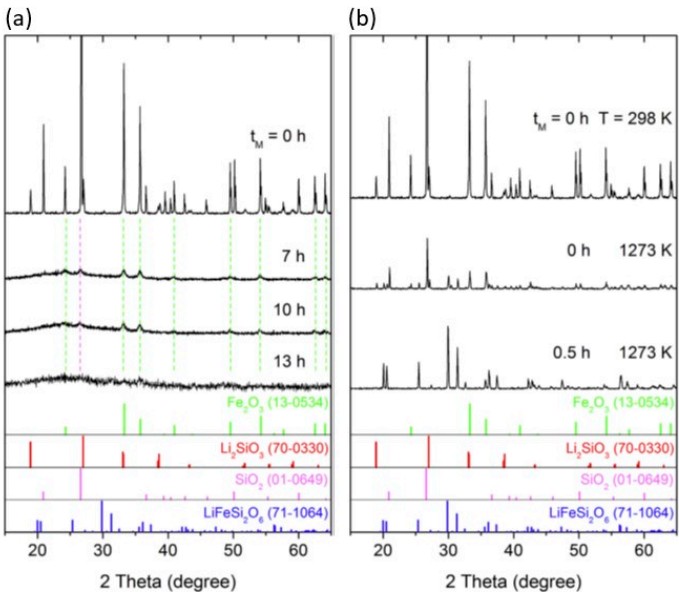

**Figure 1.** X-ray diffractograms of pyroxene $LiFeSi_2O_6$ obtained (**a**) by milling up to 13 h to obtain nanoglassy state, and (**b**) by milling only for 0.5 h and subsequently heating at 1000 °C to obtain nano-crystalline state. Reprinted with permission from [69].

Although the milling time was short, the effect of the mechanical pre-treatment is clear. When we heated the intact mixture under the same conditions, we did not obtain the same crystalline state. A clear difference between these two products was observed in the short-range ordering around Fe using Mössbauer spectroscopy. As shown in Figure 2a, the mechanosynthesis products after prolonged milling exhibited significantly higher quadrupole splitting (QS) compared to that of well-crystallized pyroxene (Figure 2b). This is a clear indication of highly distorted local, short-range atomic ordering and hence a glassy state.

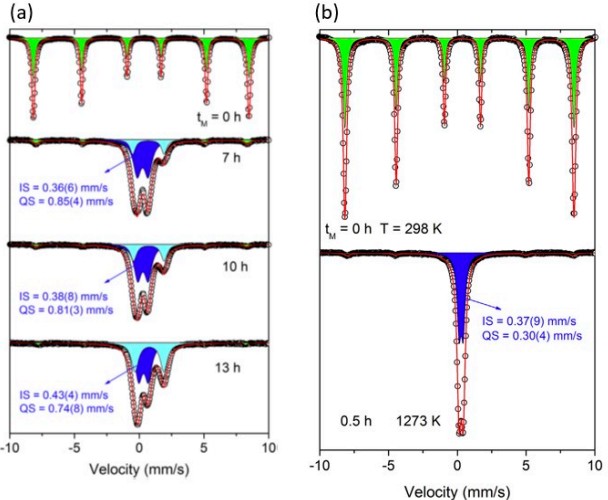

**Figure 2.** Mössbauer spectra of $LiFeSi_2O_6$ of (**a**) nanoglassy state, and (**b**) nano-crystalline state. Reprinted with permission from [69].

## 4. Spinel and Perovskite Compounds with Varying Anti-Site Disorder

An even finer structural view is atomic disorder or anti-site disorder. Traditional examples are found in ferrites, where normal and inverse spinel structures are controlled during mechanosynthesis [70,71]. Note that the products are understood as non-equilibrium states. A more complicated example can be found in perovskite. The structural control between normal and anti or reverse perovskite involves similar problems [72,73].

Double perovskites, $A_2BB'O_6$ (DPVs), are much more complicated in structure than conventional spinels. They are now under the spotlight, mainly for magneto-optical or spintronic applications [12]. Here, the specific concept of anti-site disorder (ASD) plays an important role in their attractive functional properties. The stabilization and control of ASD is one of the main issues in DPV synthesis. Baranowski et al. stated [74] that the stability of high anti-site disorder has been discussed in terms of entropy-driven cation clustering. A further step towards the control of ASD is also being studied intensively [75]. An explicit example of ASD is explained in Figure 3 for $Sr_2FeMoO_6$ (SFMO) from our recent study [20].

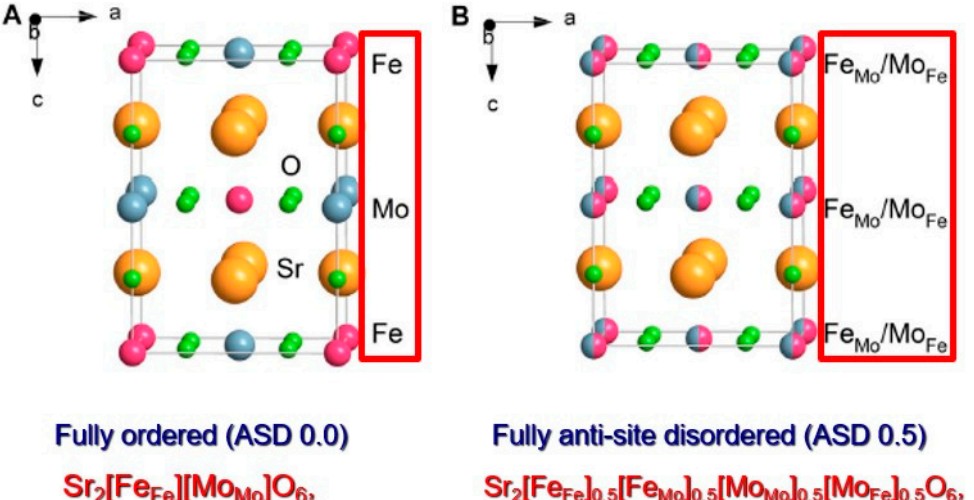

**Figure 3.** Crystalline model of $Sr2FeMoO_6$ double perovskite with (**A**) totally ordered (ASD = 0) and (**B**) fully disordered (ASD = 0.5) states. Reprinted with permission from [20].

When two transition metal atoms, Fe and Mo, occupy their reserved positions, we count the ASD as zero (Figure 3a). When they share two positions half and half, we recognize the total disorder and count ASD as 0.5 (Figure 3b).

When we ground a mixture of SrO, α-Fe, and $MoO_3$ in an exact stoichiometric ratio of SFMO in a conventional planetary mill for only 60 min, we obtained almost phase-pure SFMO, as shown in Figure 4. This was further verified by Raman lattice mode spectra, as shown in Figure 5.

However, Mössbauer spectroscopy revealed that the as-ground product was not phase-pure, even after 240 min of milling (Figure 6a). After heating the ground product at 923 K for 30 min in air, it became phase pure (Figure 6b). The average particle size was about 20 nm even after post heat treatment. The degree of anti-site disorder was close to 0.5, indicating full disorder, independently and consistently estimated by XRD, Mössbauer spectra, and magnetization data. This kind of structural state, far from equilibrium and highly disordered, has never been achieved by any previously reported method.

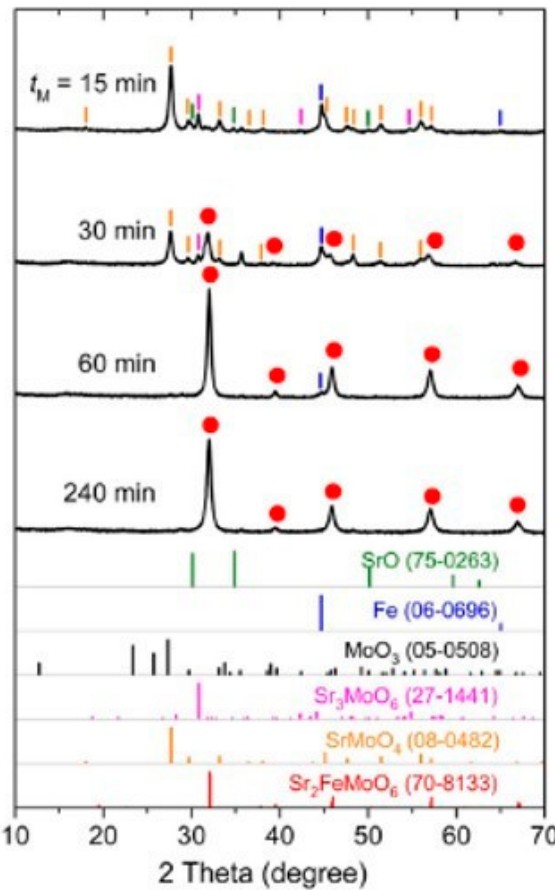

**Figure 4.** X-ray diffractograms of the milled mixtures of SrO, α-Fe, and MoO$_3$. Reprinted with permission from [20].

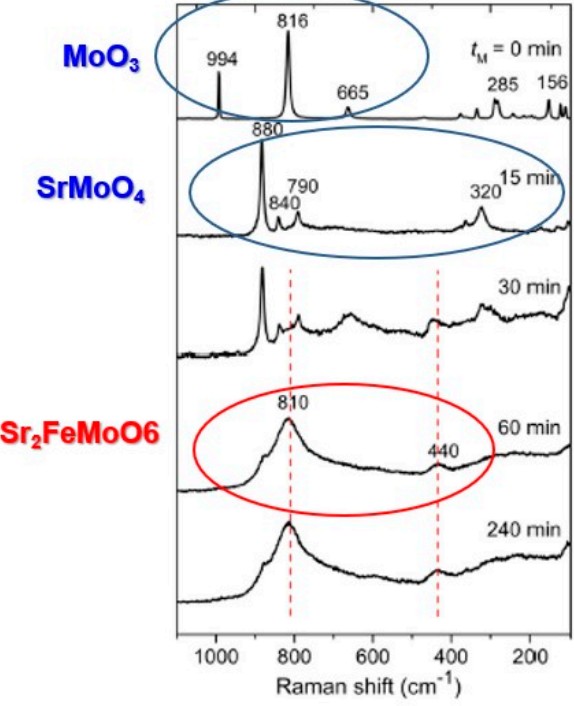

**Figure 5.** Raman lattice mode spectra of the milled mixtures of SrO, α − Fe, and MoO$_3$. Reprinted with permission from [20].

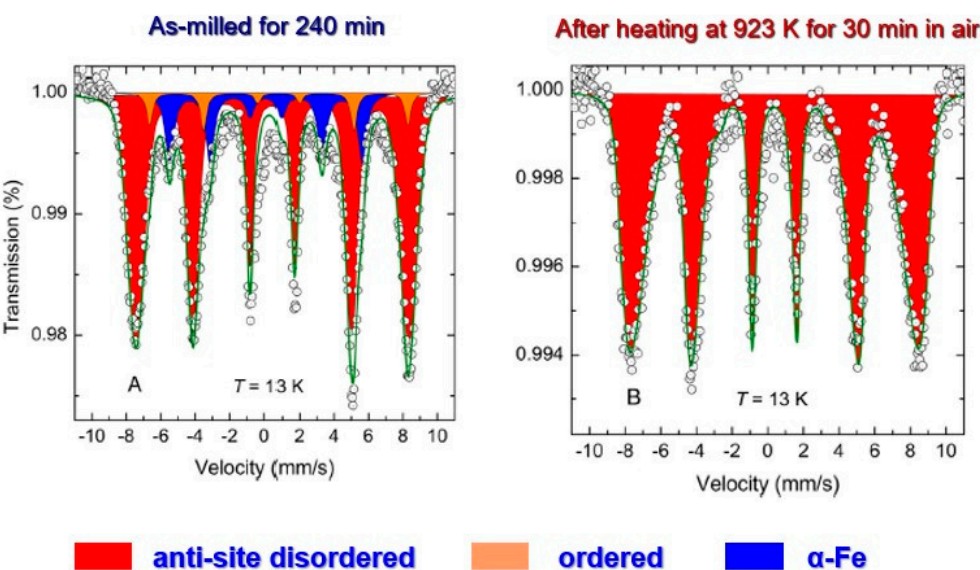

**Figure 6.** [57]Fe Mössbauer spectra at 13K of 240 min milled mixture; as—milled (**left**) and with subsequent heating at 923 K for 30 min in air (**right**). Reprinted with permission from [20].

## 5. Stabilized Molecular Dispersion in Pharmaceutic Technology

As mentioned in the introduction, the application of mechanochemistry-related technology to pharmaceutics and pharmacology is a rapidly growing and strong driving force of its industrialization. While the greener processing of organic synthesis for active materials is the mainstream of the top-running trend of mechanochemistry to pharmaceutical technology [39,76], the issue of scaling up is becoming increasingly attractive [4,41]. The traditional stream of mechanochemical technology in pharmaceutics, i.e., the amorphization of active ingredients, also needs to be re-evaluated.

In pharmaceuticals, people almost always try to make the API as amorphous as possible for the purpose of the faster dissolution, bioavailability, and efficiency of drug absorption [77]. When the API is dispersed at the molecular level, i.e., in the extremes of its amorphous state, we can expect the highest possible bioavailability. The latter states, called amorphous solid dispersion [44,45], are achieved via a mechanochemical route [78–80]. However, amorphous states are less stable than crystalline states. The stabilization of the state of molecular dispersion is therefore very important, but has been much less studied. Highly deformed, activated states are stabilized, to some extent, when such a state is fixed to neighboring substances, excipients [81–83]. They are mostly based on the choice of excipient species. However, there are other ways of thinking about how to stabilize the state of molecular dispersion with a fixed excipient. Here, mechanochemical processing plays a different role.

The amorphization and molecular dispersion of a popular anti-inflammatory drug, indomethacin (IM), was conducted. It was subjected to amorphization via a conventional melt-quench and a mechanochemical route and these were compared [78]. The crystallized IM could not be amorphized when milled alone for 180 min, as shown in Figure 7. When the IM was ground together with fumed silica, it apparently amorphized after 30 min of grinding. The state of amorphization via conventional melt quenching was also shown for comparison. In the latter case, the coexistence of fumed silica did not play a significant role.

Upon the exposure of once-amorphized compounds, amorphized drugs tend to recrystallize. When the amorphized IM was exposed via melt quenching in a well-controlled environment of 30 °C and 11% relative humidity, recrystallization became apparent after 5 days and it was completely recrystallized after 3 days, as shown in Figure 8.

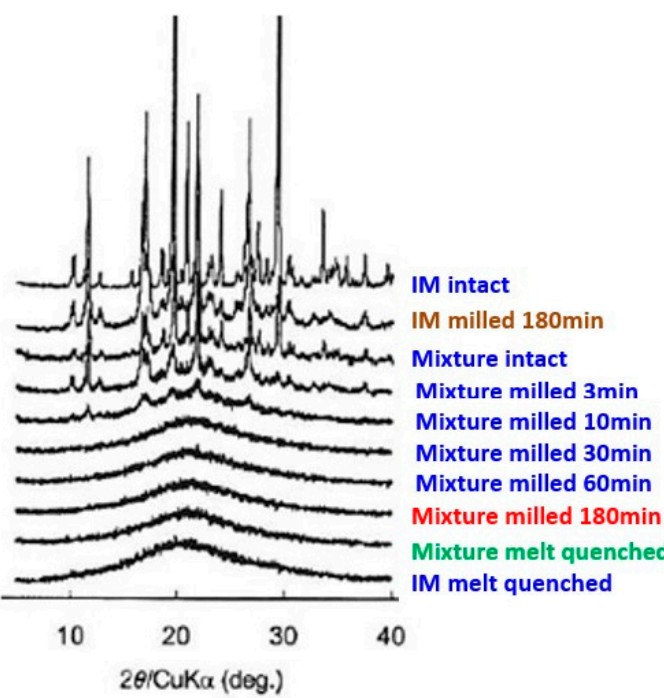

**Figure 7.** Change in XRD of IM–silica mixture by milling and melt-quenching. Reprinted with permission from [78].

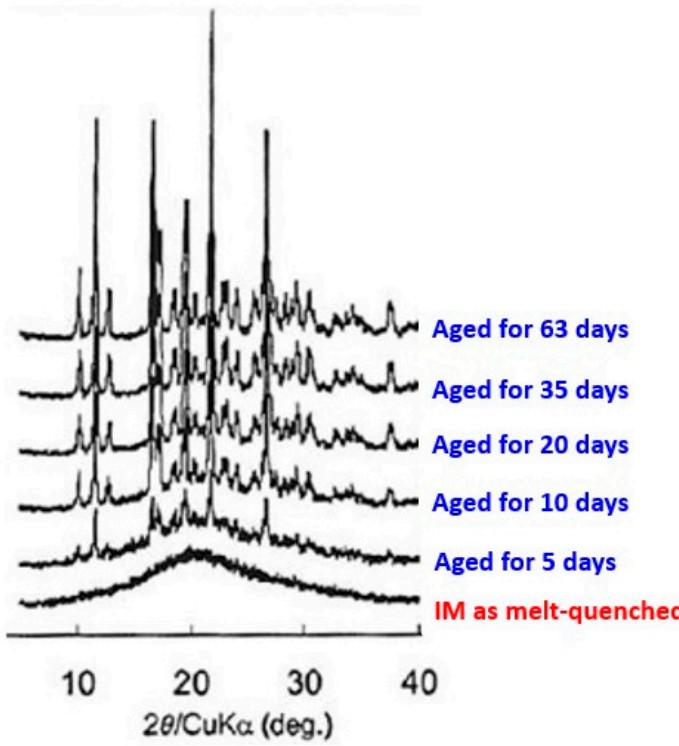

**Figure 8.** Recrystallization of melt quenched IM after aging at 30 °C and 11% relative humidity. Reprinted with permission from [78].

As shown in Figure 9, the recrystallization kinetics of the amorphized IM were very different depending on the amorphization method. The degree of crystallinity was suppressed to about 10% even after 3 months when co-milling was extended to 180 min.

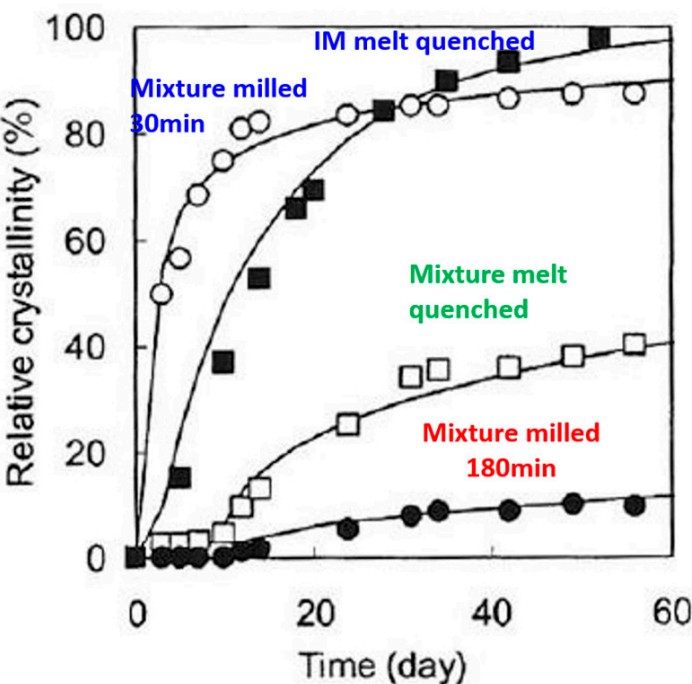

**Figure 9.** Recrystallization kinetics of amorphized IM upon aging at 30 °C and 11% relative humidity. Reprinted with permission from [78].

Unfavorable fast recrystallization is associated with the debris of the surviving crystallite region, which serves as the active site for recrystallization. A closer look is needed to elucidate the state of the specific chemical interaction between the active drag materials and the excipients, i.e., the IM and fumed silica in this case. Hydrogen bonding is the main chemical interaction between them, as shown schematically in Figure 10 [84].

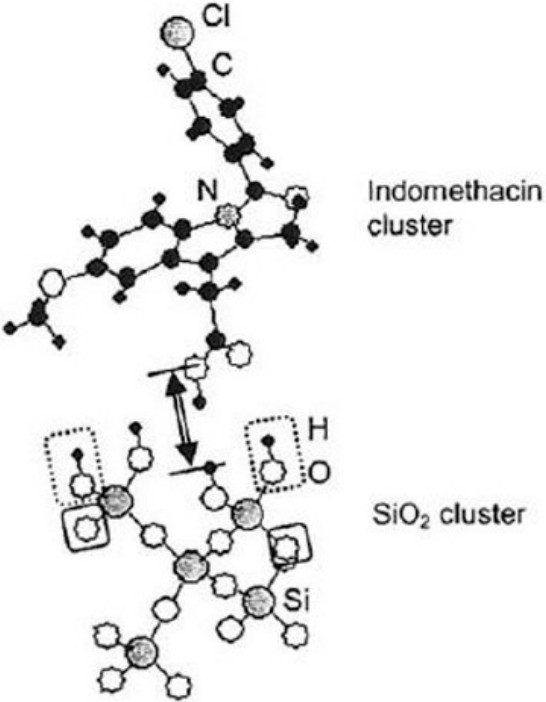

**Figure 10.** A scheme of bridging bond between IM and silica. Reprinted with permission from [84].

The interaction state also changed with the milling time of the mixture. As shown in Figure 11 [85], XPS profiles indicated the decrease in Si2p binding energy, which indicated

an increase in electron density, i.e., an apparent reduction in silicon ionic states. The difference was further confirmed by $^{13}$C-CP/MAS NMR, as shown in Figure 12 [39].

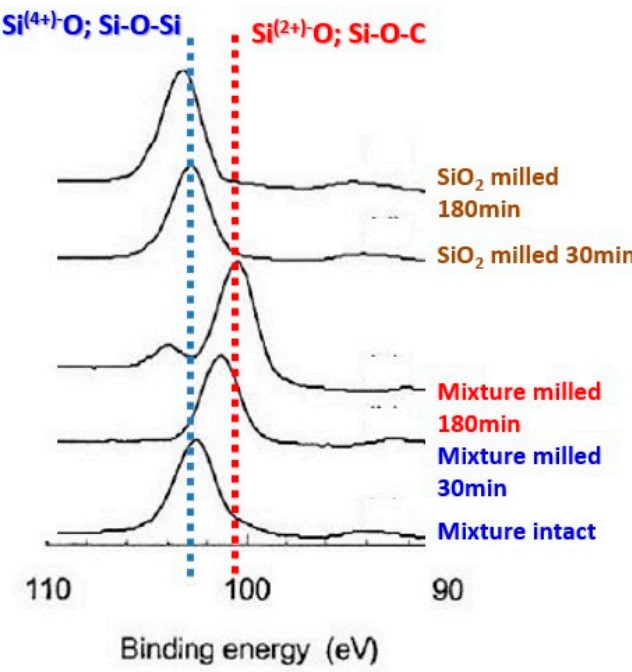

**Figure 11.** Si2p XPS spectra of various IM—silica mixtures. Reprinted with permission from [85].

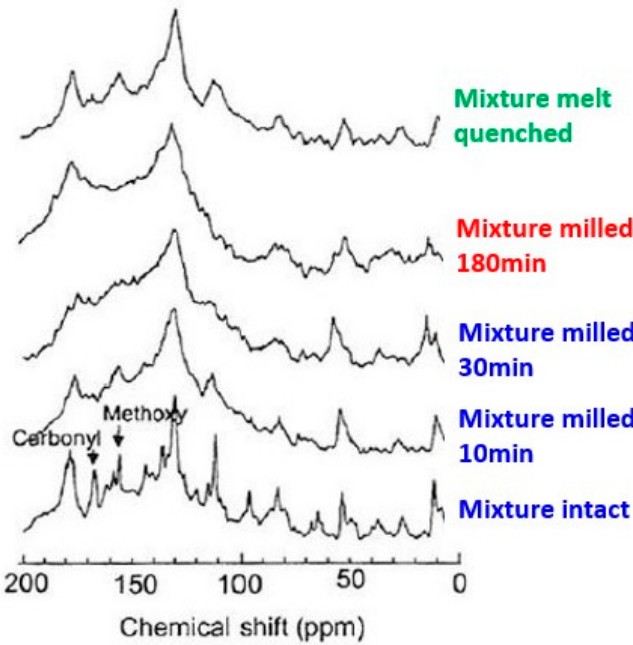

**Figure 12.** $^{13}$C-CP/MAS NMR spectra of various IM–silica mixtures. Reprinted with permission from [78].

## 6. Involvement of Liquids and Auto-Liquefaction

Mechanochemical processes are considered green, especially in organic synthesis, mainly due to the absence of solvents. On the other hand, the addition of small amounts of liquids, not always considered as solvents, often shows positive effects [2]. A change in the mechanical properties of metals with surfactants was discussed as an effect of liquids on the mechanical properties of metallic materials under the concept of Rehbinder effects [86].

The effects were interested in the machining of metallic species, where mechanochemical effects are involved [23,87]. Since most of mechanochemical processing is carried out with a milling machine, it is worthwhile to briefly compare dry and wet milling. There are several aspects related to mechanochemical effects.

The type and amount of mechanical stress are common to particle size reduction, which is the primary purpose of conventional milling. A detailed simulation study discussed the velocity dependence of the coefficient of friction, which was similar to the variation of the coefficient of friction with sliding velocity, as described using the lubrication theory [88]. The intensity of the mechanochemical effects is easily comparable to those of the conventional mechanical milling of powders. The energy efficiency of the pilot mill was discussed to obtain the power demand during wet and dry grinding processes [89]. This is also common to all processes that use grinding equipment. A more specific comparison between wet and dry grinding was made in the pharmaceutical industry and provided useful information in this area [90]. Attention has also been paid to the electrostatic charging in dry grinding. Even finer processing, such as DNA extraction from fungal spores, showing the merits of selecting dry versus wet milling, was thoroughly discussed [91].

The concept of liquid-assisted grinding (LAG), proposed by Friscic et al. [92,93], is associated with the mechanosynthesis of cocrystals. The role of liquid is far more than that of lubricant. The effects of liquid addition are particularly remarkable for cocrystals with more than two components, due to the acceleration of the inclusion of participating molecular species and a subtle difference in the phase stability with the partial dissolution of a particular constituent.

There are quite different aspects of liquid phase involvement in organic mechanosynthesis, i.e., the emergence of the liquid phase during mechanochemical synthesis starting from dry powder mixtures. An example of this is the solid–solid cocrystal formation between thymol and hexamethylenetetramine, which proceeds via the formation of a metastable binary low-melting eutectic [94,95]. The use of the in situ formed liquid phase has another advantage for solvent-insoluble solid phases, since the liquid phase appears only as an intermediate during the mechanochemical process, while the solid phase is the final product [96]. Mechanically induced reactions between triphenylphosphine and organic halides are also suspected of passing through low-temperature eutectics, e.g., triphenylphosphine-organic bromide systems during ball milling. The formation of the eutectic liquid phase may be local and transient in grinding jars. However, the participation of such a liquid phase can significantly accelerate the mechanochemical reactions [97,98]. Similar mechanochemical reactions through the participation of the eutectic liquid phase are also found for wider genres of organic synthesis, such as Diels–Alder reactions [48].

Liquid-phase intermediates are interpreted as charge transfer complexes (CTCs) in liquid or solid states. The formation of such CTCs is often attributed to the π–π interaction being dominated by the orbital overlap rather than the electron density on the molecular orbital, which is verified experimentally by the process of CTC and DFT calculations. The intensity of the π–π interaction is further influenced by the cationic properties of the functional groups [99]. One of the unique participations of liquid in mechanochemical processes is an autogenous formation of eutectic compounds, whose melting point is at around room temperature or lower [100]. These phenomena occur spontaneously and are easily applied to organic syntheses [48,95,99]. An example is exhibited in Figure 13, which shows the starting materials and final products as powders, passing through a liquid containing intermediates [48].

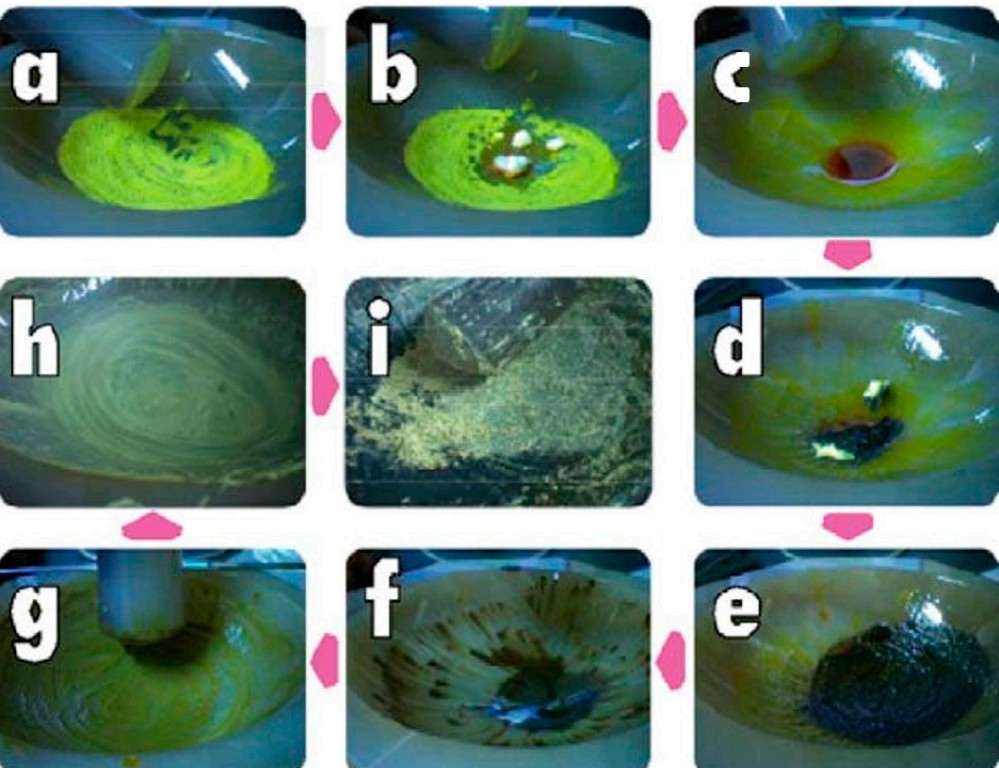

**Figure 13.** Time-resolved observation of a reaction course of eutectic Diels–Alder reaction. Crystalline powders of thymol (TM) (2 mmol), p-benzoquinone, BQ (1 mmol), and dimethylanthracene, DMA (1 mmol) were mixed in an agate mortar and pestle in an ambient condition: (**a**) quinone powder, (**b**) TM added, (**c**) autogenous fusion, (**d**) DMA added (0 min), (**e**) 1 min, (**f**) 5 min, (**g**) 10 min, (**h**) 20 min, and (**i**) reaction completed with solidification (30 min). Reprinted with permission from [48]. See details in ref [48].

## 7. Apparent Stabilization of Mechanochemical Products

The products of mechanosynthesis are categorically unstable or metastable if we follow the classical thermodynamic definition. On the other hand, all solid powdery substances are unstable or metastable because they have surfaces where the original states of coordination are disturbed. A perfect single crystal of an infinite size is the only stable solid that does not exist. In this section, relative stability or kinetic stability is discussed in connection with the products of mechanochemical processing, some of which are shown above.

As for the explicit case studies mentioned in Sections 3–5, three different mechanisms of this apparent stabilization could be demonstrated. In the case of exterior tiles, their joints play an important role. As shown in Figure 13, nanoglass materials consist of glassy nanograins surrounded by grain boundaries, just like the tiles and joints, as illustrated in Figure 14 [53]. While glassy states are always less stable than crystalline states, coexisting grain boundaries buffer the recrystallization of the glassy parts, so the nanoglass states maintain their metastability.

The increase in the anti-site disorder to its full disorder after heating cannot be fully explained on a theoretical basis. One possible explanation is the contribution of configurational entropy [74,101,102]. In fact, the concept of High Entropy Alloy (HEA) is rapidly gaining acceptance, where alloys with more than three cationic species remain stable simply because of the entropy contribution to the thermodynamics of multicomponent alloys [103]. It should be noted, however, that there are many factors involved, including differences in atomic size or the amount of deformation. This concept is also extended from alloys to complex oxides, nitrides, or other complex compounds under the concept of High Entropy Materials (HEM) [11]. A similar concept is applicable to more complex interfacial

phenomena, including biomaterials [104]. The mechanosynthesis of HEM has also been reported [58]. The stability of such products is explained by the concept of entropy-driven stabilization [101].

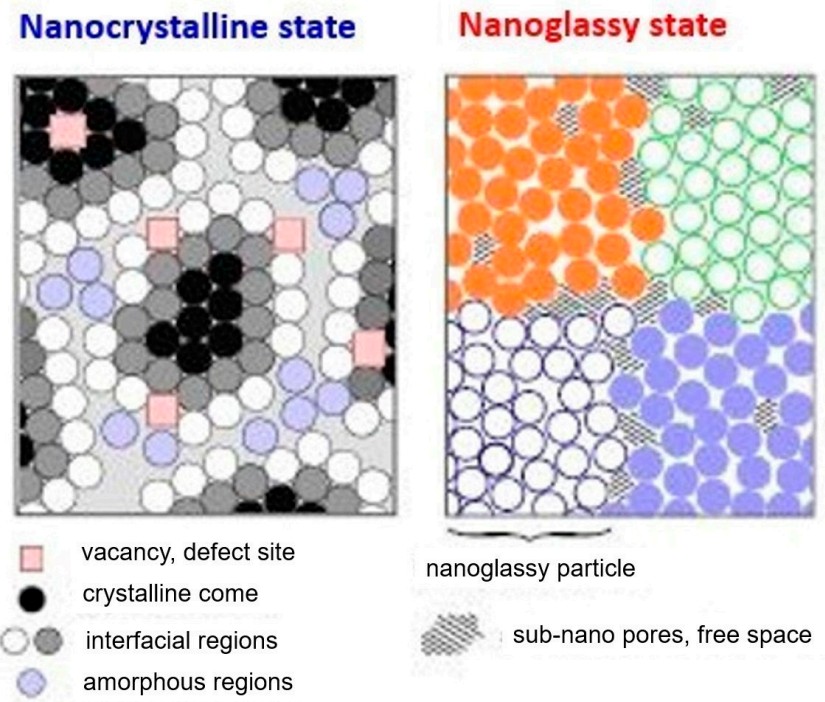

**Figure 14.** Schematic illustration of nano-crystallin (**left**) and nano-glassy (**right**) states. Reprinted with permission from [64].

In the case of molecular dispersion, the molecular crystal of a drug is gradually disintegrated under mechanical stress. Its debris is further broken down into the states of independent molecules, which inevitably interact with the coexisting excipient. The drug–excipient bond is non-covalent, typically hydrogen bonds (HBs), as extensively studied in the interest of the controlled release of drugs [105–107]. The strength of HBs varies greatly depending on the structural disorder of the excipient solid and/or coexisting OH groups or hydronium ions. This is similar to those discussed for molecular sieves, where molecules are trapped and confined in the micropores of the zeolite [108]. These molecular–substrate interactions hinder the recrystallization of molecular crystals and thus stabilize the highly active molecular dispersion state. It should also be noted that the mechanochemical process not only facilitates the formation of HBs, but also provides different options for their intensity.

## 8. Remarks for the Process Optimization

As mentioned above, there are large numbers of items in the field of mechanochemistry associated with functional nanocomposites. This section summarizes the preferential items needed for some crucial processes.

1.  Size-dependent properties

A milling device. This is used in mechanochemical processing is designed for the size reduction in the powder particle. It is therefore usual that most mechanical activation is accompanied by a decrease in the average particle size, often down to a nanometer regime. Most of the physical properties of the solid are size-dependent. Therefore, we always have to pay attention to the property change in the properties. It is primarily important to the change in the plasticity. Local plastic deformation, which occurs automatically, even in brittle materials for local areas or in small enough particles [109], is beneficial for further migration into the interior of the new granules.

Particle size dependence is further extended to ionic diffusivity [110]. Diffusion constants are considered to be independent of particle size. However, when diffusive migration passes through low-dimensional atomic migration channels, it depends on particle size, due to the change in the distance of the point defect, which limits the diffusivity, as shown in Figure 15 [110]. This could be associated with the changes during mechanochemical treatment.

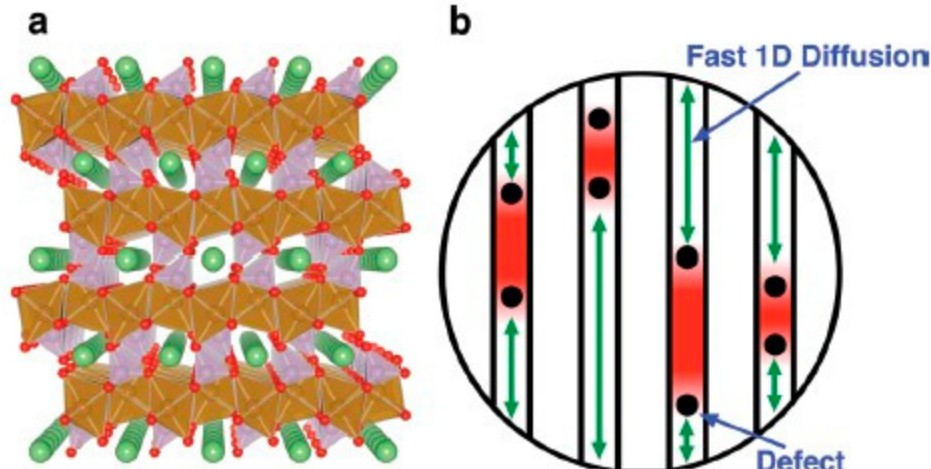

**Figure 15.** Schematic illustration of nano-crystallin (a) and nano-glassy (b) states. Reprinted with permission from [110].

A new bridging bond across the surfaces of the neighboring dissimilar particles is preceded by charge transfer, similar to contact charging. For successful transfer, the jump distance should be small enough. The formation of these new bridging bonds is verified by XPS as a change in the electron density distribution and by mid-Raman spectra as the formation of a new lattice mode peak. Ionic conductivity is of the utmost importance for the materials for the storage and conversion of energy. It is important to recognize that the ionic diffusion across the particle boundary has a definite similarity to the ionic migration across the grain boundary in polycrystalline materials [111].

2.    Precision control of short-range disorder

The lattice disorder in materials is strongly associated with their physical properties, while the effects of long-range ordering, as determined by XRD, are mainly due to the mechanical properties in terms of dislocation [112]. A general relationship has been discussed in the case of conjugated polymers [113], but not for the materials with more complicated structures. Various methods have been proposed to control the short-range disorder, and the concept of disorder engineering will be increasingly spotlighted [114]. A similar control of the short-range disorder can also be achieved by mechanical stressing [115].

3.    Sustainability and affordability

The mechanochemical processes consume a lot of energy. Therefore, it is important to keep these processes as mild and short as possible. Since most of them are solvent-free, the energy-intensive thermal drying process can be saved. This is, by itself, very sustainable. The discussion here is concentrated on how to make the mechanochemical process more sustainable. The use of additives such as used grinding aids traditionally used in the cement industry is also effective for mechanical activation. Nothing is more sustainable than when a reaction proceeds spontaneously and automatically. Autogenous eutectic formation with its liquefaction fulfills this condition [48,95,99]. Scaling up is a new hope for the industrial application of mechanochemical processes. Organic synthesis was scaled up by using an extruder. This technology is particularly interesting for pharmaceutical processes [116,117].

Replacing the starting material with natural resources such as biomass in mechanochemical processes is entirely different, but also an important viewpoint for the modification of the process toward sustainability. Indeed, many petrochemical processes could be replaced by biorefinery [118–120].

## 9. Concluding Remarks and Outlook

The microscopic transport of chemical species in and across particulate solids is promoted under applied external mechanical stress. The control of the short- and long-range ordering in nanostructured materials via a mechanochemical process will become widely available in an affordable manner. With a step-by-step understanding of the actual processes involved, these processing technologies will become even more affordable for producing the products we need today and in the near future in a more environmentally benign manner. As briefly summarized in Section 8, process optimization in mechanochemistry can be achieved by refining the microscopic elementary steps studied through a series of well-targeted characterizations. With such refinement, we can anticipate a bright future for the application of mechanochemical principles in line with the concept of the SDGs in a truly affordable manner.

**Funding:** This research received no external funding.

**Acknowledgments:** The author sincerely thanks the coworkers appeared in the case studies introduced, among others, Tomoyuki Watanabe (Daiichi Sankyo Company, Limited), Vladimir Šepelák (Institute of Nanotechnology, Karlsruhe Institute of Technology), and Erika Tóthová (Institute of Geotechnics, SAS).

**Conflicts of Interest:** The authors declare no conflict of interest.

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
