# Peer review of "The Optimization of Mechanochemical Processes toward Functional Nanocomposite Materials"

_2674-0516, doi:10.3390/powders2030041_

Round 1

Reviewer 1 Report

 1. “The present review focuses on some new concepts focused on those at organic-inorganic powder particle boundaries, aiming at optimizing mechanochemical processes for nanocomposites in various fields including electromagnetic functional materials and pharmaceutical technology”.

-This statement is hardly related to the most of really interesting facts and experiments described in this brief review. The most of them deal with truly inorganic or truly organic materials (pharmaceuticals), not with organic/inorganic interfaces in the corresponding composites. It would be reasonable to modify the objectives of this review in order to make it closer to the actual content of this paper.

 2. “The present author, together with his colleagues, has demonstrated some examples of this possibility [63, 64].”

- Ref. 64 is written by other authors and deal with metal-organic frameworks, not with nanoglass materials.

“Traditional examples are found in ferrites, where normal and inverse spinel structure are controlled during mechanosynthesis [65-67].”

- Ref. 67 is not related to spinel ferrites.

It seems that the list of references of this paper needs more detailed revision.

 3. “On the other hand, the addition of small amounts of liquids, not always considered as solvents, often shows positive effects [2].”

- In fact, far before these studies the similar effects were observed and studied by Rehbinder et al.

Author Response

Reviewer 1

Reviewer 1

Comments and Suggestions for Authors

  1. “The present review focuses on some new concepts focused on those at organic-inorganic powder particle boundaries, aiming at optimizing mechanochemical processes for nanocomposites in various fields including electromagnetic functional materials and pharmaceutical technology”.

-This statement is hardly related to the most of really interesting facts and experiments described in this brief review. The most of them deal with truly inorganic or truly organic materials (pharmaceuticals), not with organic/inorganic interfaces in the corresponding composites. It would be reasonable to modify the objectives of this review in order to make it closer to the actual content of this paper.

I have explicitly defined the objectives of the present review at the end of the Introduction, i.e., "The objective of this review is to offer a guideline for the selection of preferential elements for various goals of mechanochemical operations, i.e., (a) formation of new bridging bonds by charge transfer; (b) precision control of lattice disorder and imperfection; (c) sustainability and affordability. They are listed in the new section "8. Notes on Process Optimization".

  1. “The present author, together with his colleagues, has demonstrated some examples of this possibility [63, 64].”

- Ref. 64 is written by other authors and deal with metal-organic frameworks, not with nanoglass materials.

Ref 64 was deleted

“Traditional examples are found in ferrites, where normal and inverse spinel structure are controlled during mechanosynthesis [65-67].”

- Ref. 67 is not related to spinel ferrites.

Ref 67 was deleted

It seems that the list of references of this paper needs more detailed revision.

I have reexamined the reference list and revised.

  1. “On the other hand, the addition of small amounts of liquids, not always considered as solvents, often shows positive effects [2].”

- In fact, far before these studies the similar effects were observed and studied by Rehbinder et al.

I have inserted the items related to the Rehbinder effect by introducing new references, i.e., Refs [32-34].

Reviewer 2 Report

This manuscript deals with the Optimization of mechanochemical processes toward functional nanocomposite materials. The findings of the study are well-presented. But in its current form, I recommend the authors to revise and resubmit their manuscript.

1.     The provided keywords are so common words of the matter. Please select more specialized keywords regarding the issue of the manuscript.

2.     Please check the manuscript for any grammatical and dictation errors.

3.     The manuscript contains an elaborate literature review, but definitions of the key concepts are needed in the introduction. Please provide a comprehensive study of previously reported literature of similar studies about mechanochemical. Provided information on the mechanochemical study is not sufficient and informative. Please add more details to the introduction.

4.     Please ensure that the topics mentioned in the introduction are followed seamlessly.

5.     Please define abbreviations used in the text only once and consistently follow the same format throughout the entire text.

6.     There are issues with reference citation, for example, one sentence in line 290: “The difference was further confirmed by 13C-CP/MAS NMR as shown in the figure. 12” Actually, reference 39 does not have the same figure and experiment as figure 12.

7.     Please improve the quality of the figures. Figure 13 is difficult to read and should be changed.

8.     Please note that third-party copyrighted material reproduced in your paper should as a general rule be cleared for use by the rights holders or please provide the complete statement of image permissions if approved by the responsible author.

9.     The manuscript draws on impressive data, as described in the methodology. However, the wealth of data does not come across in the analysis. I believe several additional tables and figures can improve the data and quality of the manuscript.

10. Most of the images used in the text are from previous studies conducted by the author of the article. However, it is recommended to also include images from other researchers or authors to provide a diverse range of perspectives and sources.

11. Please provide further clarification on the outcome.

minor editing of English language required

Author Response

Reviewer 2

Comments and Suggestions for Authors

This manuscript deals with the Optimization of mechanochemical processes toward functional nanocomposite materials. The findings of the study are well-presented. But in its current form, I recommend the authors to revise and resubmit their manuscript.

  1. The provided keywords are so common words of the matter. Please select more specialized keywords regarding the issue of the manuscript.

Keywords are replaced to meet the contents

  1. Please check the manuscript for any grammatical and dictation errors.

English was brushed up by referring to the comments of my fellow researcher, whose native language is English.

  1. The manuscript contains an elaborate literature review, but definitions of the key concepts are needed in the introduction. Please provide a comprehensive study of previously reported literature of similar studies about mechanochemical. Provided information on the mechanochemical study is not sufficient and informative. Please add more details to the introduction.

I have explicitly refined the objectives of the present review at the end of the Introduction, i.e., "The objective of this review is to offer a guideline for the selection of preferential elements for various goals of mechanochemical operations, i.e., (a) formation of new bridging bonds by charge transfer; (b) precision control of lattice order and imperfection; (c) sustainability and affordability. They are listed in the new section "8. Notes on Process Optimization".

  1. Please ensure that the topics mentioned in the introduction are followed seamlessly.

I have extensively revised the Introduction. At the end, additional paragraph of the objectives are added, i.e., “The objective of this review is to offer a guideline of selecting preferential items for various goals of mechanochemical operations, i.e.,  (a) new bridging bond formation by charge transfer; (b) precision control of lattice order and imperfection; (c) sustainability and affordability”.

  1. Please define abbreviations used in the text only once and consistently follow the same format throughout the entire text.

Collected as suggested

  1. There are issues with reference citation, for example, one sentence in line 290: “The difference was further confirmed by 13C-CP/MAS NMR as shown in the figure. 12” Actually, reference 39 does not have the same figure and experiment as figure 12.

The reference number was corrected.

  1. Please improve the quality of the figures. Figure 13 is difficult to read and should be changed.

The quality was imporuved

  1. Please note that third-party copyrighted material reproduced in your paper should as a general rule be cleared for use by the rights holders or please provide the complete statement of image permissions if approved by the responsible author.

All of them were reproduced under permission. It was stated in the caption, and the permission protocol is attached.

  1. The manuscript draws on impressive data, as described in the methodology. However, the wealth of data does not come across in the analysis. I believe several additional tables and figures can improve the data and quality of the manuscript.

As mentioned in the reply to the comment 3, some new concepts are summarized in the newly inserted section 8 with substantial number of new references. An additional figure was added as Fig. 13 (in the revised version) under permission.

  1. Most of the images used in the text are from previous studies conducted by the author of the article. However, it is recommended to also include images from other researchers or authors to provide a diverse range of perspectives and sources.

In the newly added section 8, Fig. 15 was added, introducing the size-dependent diffusion pathways by external authors from ref [112].

  1. Please provide further clarification on the outcome.

I have made the objectives of the review clear as mentioned at the end of the Introduction. A new section, i.e., 8. Remarks for the process optimization.

Round 2

Reviewer 2 Report

Thank you for the revision.

Author Response

Thank you again for your time to review my revised version.

Bes regards

Mamoru Senna